# Causal relationship between periodontal disease-related phenotype and knee osteoarthritis: A two-sample mendelian randomization analysis

**Longqiang Shen, Di Niu, Gang Deng** *

Institute of Blood Transfusion at Ningbo Central Blood Station, Ningbo, Zhejiang, China

* dgflying@126.com

## Abstract

### Objective

This study aimed to explore the bidirectional causal relationship between periodontal disease-related phenotype (PDRP) and knee osteoarthritis (KOA) in a European population using a two-sample Mendelian Randomization (MR) approach.

### Methods

We leveraged publicly available GWAS summary statistics for PDRP (n = 975) and KOA (n = 403,124), assessing their roles as both exposures and outcomes. Our comprehensive MR analysis employed various methods, including inverse variance weighting (IVW), weighted median, Egger regression, simple mode, and weighted mode, to enhance the robustness of our findings. To ensure the reliability of our instrumental variables, we implemented a rigorous screening process based on p-values and F-values, utilized Phenoscanner to investigate potential confounders, and conducted sensitivity analyses.

### Results

Our analysis identified five SNPs associated with PDRP and three SNPs with KOA, all surpassing the genome-wide significance threshold, as instrumental variables. The IVW method demonstrated a significant causal relationship from PDRP to KOA (beta = 0.013, SE = 0.007, P = 0.035), without evidence of directional pleiotropy (MR-Egger regression intercept = 0.021, P = 0.706). No support was found for reverse causality from KOA to PDRP, as further MR analyses yielded non-significant P-values. Additionally, funnel plots and Cochran's Q test detected no significant heterogeneity or directional pleiotropy, confirming the robustness of our results. In multivariate analysis, when considering smoking, alcohol consumption, BMI collectively no direct causal relationship between KOA and PDRP. Conversely, smoking and higher BMI were independently associated with an increased risk of KOA.

**Data Availability Statement:** Data files are available from the GWAS Catalog and described in manuscript. the information about papers included

as below. Genome-wide association study of biologically informed periodontal complex traits offers novel insights into the genetic basis of periodontal disease. Offenbacher S, Divaris K, Barros SP, Moss KL, Marchesan JT, Morelli T, Zhang S, Kim S, Sun L, Beck JD, Laudes M, Munz M, Schaefer AS, North KE. Offenbacher S, et al. Hum Mol Genet. 2016 May 15;25(10):2113-2129. doi: 10.1093/hmg/ddw069. Epub 2016 Mar 8. Hum Mol Genet. 2016. PMID: 26962152 Free PMC article.

———————————————————

Identification of new therapeutic targets for osteoarthritis through genome-wide analyses of UK Biobank data. Tachmazidou I, Hatzikotoulas K, Southam L, Esparza-Gordillo J, Haberland V, Zheng J, Johnson T, Koprulu M, Zengini E, Steinberg J, Wilkinson JM, Bhatnagar S, Hoffman JD, Buchan N, Süveges D; arcOGEN Consortium; Yerges-Armstrong L, Smith GD, Gaunt TR, Scott RA, McCarthy LC, Zeggini E. Tachmazidou I, et al. Nat Genet. 2019 Feb;51(2):230-236. doi: 10.1038/s41588-018-0327-1. Epub 2019 Jan 21. Nat Genet. 2019. PMID: 30664745 Free PMC article.

**Funding:** This study was supported by grants from Project of Medical Science and Technology Planning Project of Ningbo (Grant No. 2021Y27). The funder of the project is Longqiang shen, and the main contribution of this paper is to write the manuscript and organize and analyze the data.

**Competing interests:** The authors have declared that no competing interests exist.

## Conclusion

In conclusion, our analysis revealed no direct causal relationship from KOA to PDRP. However, a causal relationship from PDRP to KOA was observed. Notably, when adjusting for potential confounders like smoking, alcohol intake, and BMI, both the causal connection from PDRP to KOA and the inverse relationship were not substantiated.

## Introduction

Knee osteoarthritis (KOA) is a degenerative condition characterized by the progressive deterioration of joint cartilage, leading to pain, reduced mobility, and, in severe cases, may necessitate total joint replacement as a treatment option. Characterized by a high prevalence that increases with age, KOA is marked by disease progression and limited joint function, imposing substantial socioeconomic burdens. The incidence of KOA in European populations is estimated at approximately 150 cases per 100,000 person-years, with a notable increase to 250 cases per 100,000 person-years among individuals aged 60 years and older, highlighting the pronounced burden of this condition in older demographics [1]. Since the mid-20th century, the incidence of KOA has continually increased. This trend is closely linked with various factors including obesity, age, sex, and dietary habits [2,3]. Additionally, Global Burden of Disease (GBD) studies have detailed the profound impact of KOA on health and socioeconomic aspects, particularly emphasizing its contribution to disability-adjusted life years (DALYs), thereby highlighting the significant implications of the condition [4,5].

Periodontal disease (PD), an inflammatory condition precipitated by oral bacteria, leads to symptoms such as periodontal swelling, bone loss, and tooth loss [6]. Chronic periodontitis (CP), a prevalent form of PD, ranks as the sixth most common inflammatory disease globally, affecting 20–50% of the population globally [7]. Recent advancements have allowed for the refinement of the periodontal disease-related phenotype (PDRP) through the inclusion of clinical data on pathogen levels and local inflammatory responses, further characterized by periodontal complex traits (PCTs) through principal component analysis [8].

Emerging evidence suggests a complex interplay between PD and chronic diseases, with studies indicating a relationship between periodontitis and the pathogenesis of radiological KOA [9]. Interestingly, patients with KOA exhibit a 2–3 folds increased risk of developing periodontitis compared to healthy controls, independent of age, sex, occupation, alcohol consumption, body mass index, or disease severity [10]. Conversely, periodontitis has been linked to an elevated risk of developing severe KOA, necessitating total knee replacement, suggesting a bidirectional relationship between these conditions [11]. Yet, to date, no Mendelian randomization (MR) studies have explored the causal dynamics between KOA and PDRP.

Mendelian randomization leverages genetic variations as instrumental variables (IVs) to infer causal relationships between traits, circumventing the limitations of reverse causality and confounding inherent to observational studies. This approach rests on three core assumptions: (a) the genetic IVs are strongly associated with the exposure; (b) the genetic IVs are not associated with confounders of the exposure-outcome relationship; and (c) the genetic IVs influence the outcome exclusively through the exposure, excluding alternative pathways (Fig 1). By exploiting the random allocation of genetic variants during gamete formation, in accordance with Mendel's laws, MR analyses can provide more robust evidence for causal inferences, circumventing key limitations of observational studies, such as reverse causation and residual confounding. Employing a bidirectional two-sample MR design, this study aims to investigate

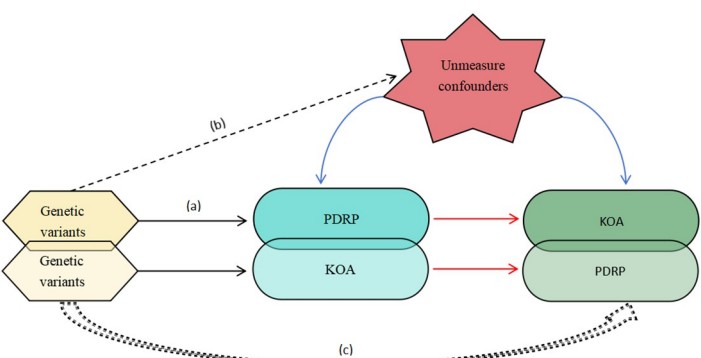

**Fig 1. A bidirectional two-samples MR model in this study.**

the potential reciprocal relationship between periodontal disease phenotype and knee osteoarthritis, contributing to the understanding of their causal interconnections.

## Materials and methods

The data utilized in this study were extracted from previously published papers [8,12], with the original research receiving approval from the Ethics Committee. This study encompasses data on a merged sample of 404,099 participants, with details on the specific GWAS samples presented in Table 1.

### Data sources

In our study, we conducted a systematic search for Genome-Wide Association Study (GWAS) summary statistics by leveraging the comprehensive GWAS database repository (https://gwas.mrcieu.ac.uk/), which compiles data from a multitude of genome-wide association studies [13,14]. Our primary dataset was derived from the Dental Atherosclerosis Risk in Communities (Dental ARIC) Study, comprising a cohort of 4910 individuals of Northern European descent, selected from four distinct communities within the United States. This cohort provided us with both genotype and clinical phenotype data. To assess the inflammatory response, gingival crevicular fluid (GCF) levels of interleukin-1β (IL-1β) were measured across four gingival sites per participant, with independent assays being conducted to calculate the mean GCF-IL1® levels for each individual. Furthermore, to evaluate the oral microbiome's composition, levels of eight periodontal pathogens were quantified using microbe-specific DNA probes in a randomly selected subset of 975 individuals from the Dental ARIC cohort, ensuring all selected subjects were of European ancestry. When potential correlation exists between the SNP under analysis and confounding factors, a multivariable Mendelian Randomization (MVMR) analysis is employed to explore the impact of these confounding factors on the

**Table 1. Details of the genome-wide association studies and datasets used in our analyses.**

| ID | Exposure or outcome | Sample size | Number of SNPs | Ancestry | Links for data download | PMID |
|---|---|---|---|---|---|---|
| ebi-a-GCST003484 | Periodontal disease-related phenotype | 975 | 2,077,804 | European | https://gwas.mrcieu.ac.uk/datasets/ebi-a-GCST003484/ | 26962152 |
| ebi-a-GCST007090 | Knee osteoarthritis | 403124 | 29,999,696 | European | https://gwas.mrcieu.ac.uk/datasets/ebi-a-GCST007090/ | 30664745 |

results of MR analysis. In this study, MVMR and TwoSampleMR packages were utilized for multivariable Mendelian Randomization analysis. Three confounding factors, namely Smoking status (id:ebi-a-GCST90029014 [15]), Alcohol consumption (id:ieu-a-1283 [16]), and Body mass index (BMI) (id:ukb-b-19953 [17]), were incorporated to investigate their effects on the bidirectional Mendelian Randomization analysis results for KOA and PDRP. Data for the three confounding factors were sourced from Open GWAS: IEU OpenGWAS project (mrcieu. ac.uk). This approach enables a comprehensive examination of the potential influence of these confounding factors on the observed causal relationships between genetic variants and the studied traits.

## Selection of instrumental variables

For SNPs achieving a genome-wide significance threshold (P-value $\leq$ 5.00E-08), we extracted aggregate statistical data (β coefficient and standard error) for five single nucleotide polymorphisms (SNPs) associated with PDRP and three SNPs associated with KOA. These SNPs served as IVs for the GWAS analyses of PDRP and KOA, as well as the potential confounding factors (smoking, alcohol consumption and BMI). The strength of the instrumental variables was assessed using the F-statistic, with an F > 10 indicating the absence of bias due to weak instrumental variables. To address potential confounders, we employed Phenoscanner, a comprehensive tool, to scrutinize the instrumental variables for any associations with known confounding factors. This crucial step helped us eliminate variables that might introduce bias into our analysis, ensuring that our instrumental variables were not confounded with outcomes unrelated to the exposure of interest.

## Statistical analysis

Our MR analysis was structured to explore the complex interplay between KOA and PDRP, initiating with a bidirectional assessment to elucidate the potential causal relationships between KOA and PDRP. To advance this analysis, we employed a multivariable MR approach, incorporating confounding factors such as smoking status, alcohol consumption, and BMI alongside KOA to assess their collective impact on PDRP. This method allowed us to examine whether the inclusion of these additional variables would alter the observed association between KOA and PDRP. Subsequently, we reversed the direction of our analysis by evaluating the influence of PDRP, in conjunction with the aforementioned lifestyle and physiological factors, on the risk of developing KOA.

 Employing a two-sample MR design allowed us to dissect the causal relationship between PDRP and KOA, merging data from various GWAS and examining their bidirectional causality. The Inverse Variance Weighting (IVW) method was used to calculate the Wald ratio for each SNP, providing a consistent estimate of the causal effect of exposure on the outcome, assuming each genetic variant satisfactorily met IV criteria.

 To address potential heterogeneity and ensure robustness, we applied several analytical methods, including MR-Egger, Weighted Median, Simple Mode, and Weighted Mode. MR-Egger regression facilitated the estimation of the causal effect (slope) and the potential average effect of pleiotropy across the genetic variants (intercept). The Weighted Median estimate provided a median-based estimate of the causal effect, offering additional robustness. Heterogeneity among SNPs was assessed using Cochran's Q statistic. Cochran's Q test was applied to test the heterogeneity of the IVs, and significant heterogeneity was considered to exist when p < 0.05. If heterogeneity was detected among the IVs, random-effects IVW was used; otherwise, fixed-effects IVW was applied. A "leave-one-out" analysis was also conducted to evaluate the influence of individual SNPs on the observed causal associations.

Statistical significance was determined at p < 0.05. All MR analyses were performed using R software (Version 4.3.1).

## Results

Following the exclusion of instrumental variables affected by linkage disequilibrium, we identified five instrumental variables for periodontal disease-related phenotype PDRP (Fig 2) and three for KOA (Fig 3), with F-values of the selected SNPs ranging from 29 to 40, indicating robust instrument strength.

Our analysis revealed a detrimental causal relationship between PDRP and KOA, as demonstrated by the IVW method (OR = 1.014, 95% Confidence Interval (CI) [1.001, 1.027] P = 0.035). Similar effect patterns were observed across other analytical methods, including the Weighted Median, MR Egger's method, Simple Mode, and Weighted Mode analysis (Table 2,

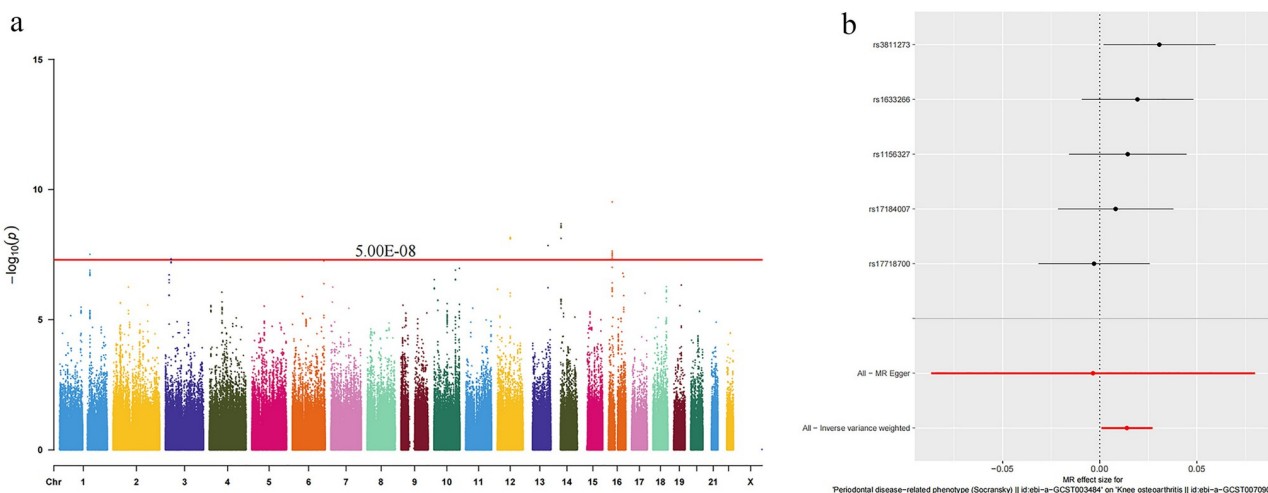

**Fig 2. Manhattan histogram of IVs for periodontal disease-related phenotype (a) and MR analysis (b).**

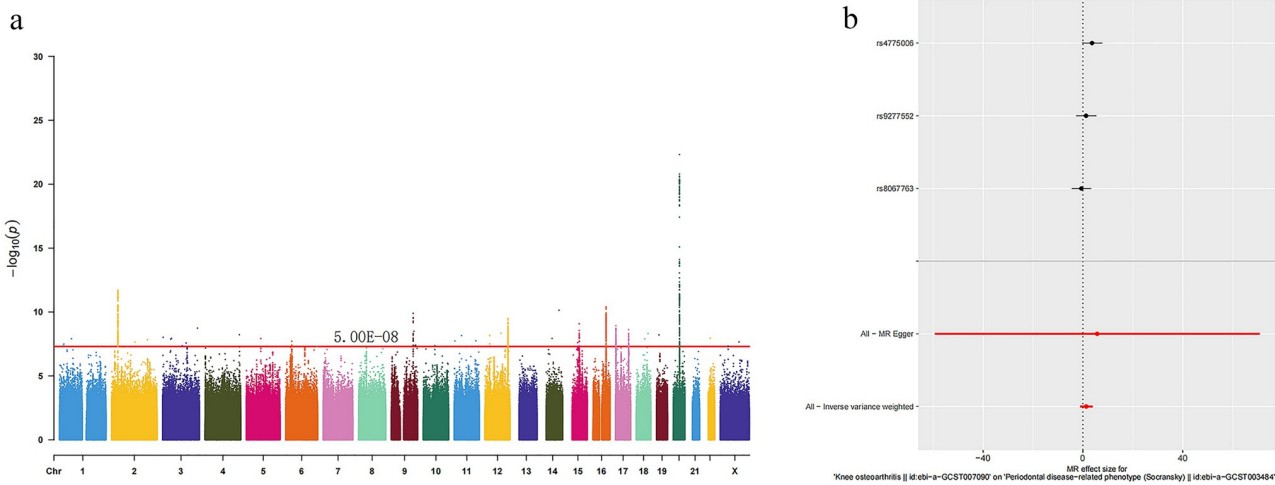

**Fig 3. Manhattan histogram of IVs for knee osteoarthritis (a) and MR analysis (b).**

**Table 2. MR estimates from each method of assessing the causal effect of PDRP on the risk of KOA.**

| Method | SNP | OR | SE | Pval | 95%CI of OR | |
|---|---|---|---|---|---|---|
| MR Egger | 5 | 0.997 | 0.042 | 0.940 | 0.917 | 1.083 |
| Weighted median | 5 | 1.014 | 0.009 | 0.116 | 0.997 | 1.032 |
| Inverse variance weighted | 5 | 1.014 | 0.007 | 0.035 | 1.000 | 1.027 |
| Simple mode | 5 | 1.015 | 0.012 | 0.280 | 0.992 | 1.038 |
| Weighted mode | 5 | 1.015 | 0.013 | 0.312 | 0.990 | 1.040 |

Fig 4), with all five SNPs meeting a less stringent statistical threshold. The MR-PRESSO test did not identify any outliers among these SNPs. Furthermore, sensitivity analyses excluding each SNP in turn confirmed that no single SNP disproportionately influenced the results, indicating a consistent positive association between PDRP and KOA (Fig 5). The Q statistic for heterogeneity among instrumental variables showed no significant heterogeneity (Q = 2.945; P = 0.567) (Table 3). Additionally, the MR Egger intercept test indicated no evidence of horizontal pleiotropy (Intercept = 0.021; Standard Error = 0.05; P = 0.706). In the PhenoScanner database, six instrumental SNPs showed no association with KOA.

Conversely, our findings did not support a causal effect of KOA on PDRP as demonstrated by the IVW method (OR = 3.846, 95% CI [0.336, 44.046], P = 0.279). The Weighted Median and other analytical methods yielded similar non-significant effects (Table 4), and evidence of horizontal pleiotropy was absent (MR Egger regression intercept = -0.26, SE = 2, P = 0.917). The Q statistic for heterogeneity among instrumental variables showed no significant heterogeneity (Q = 2.370; P = 0.306) (Table 3).

Following an initial exploration, our multivariable MR analysis delved deeply into the causal relationships between KOA and multiple exposures, including periodontal disease-

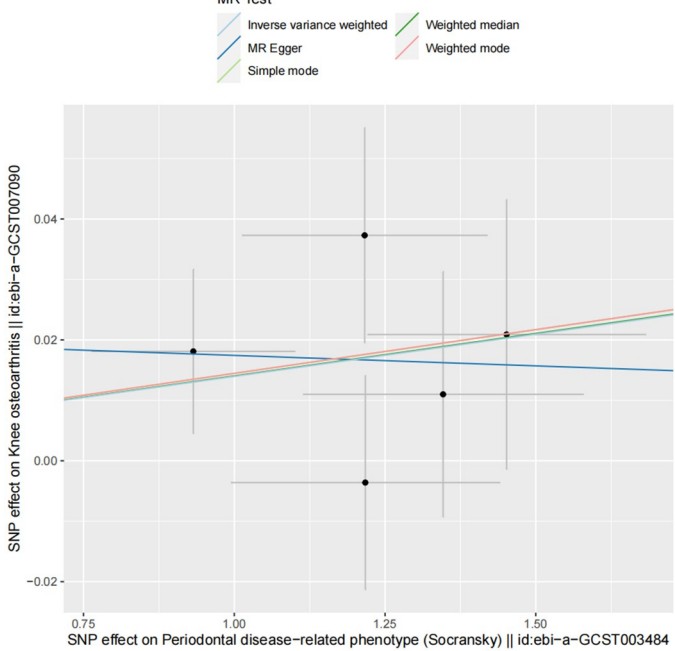

**Fig 4. MR scatter from each method of assessing the causal effect of PDRP on the risk of KOA.**

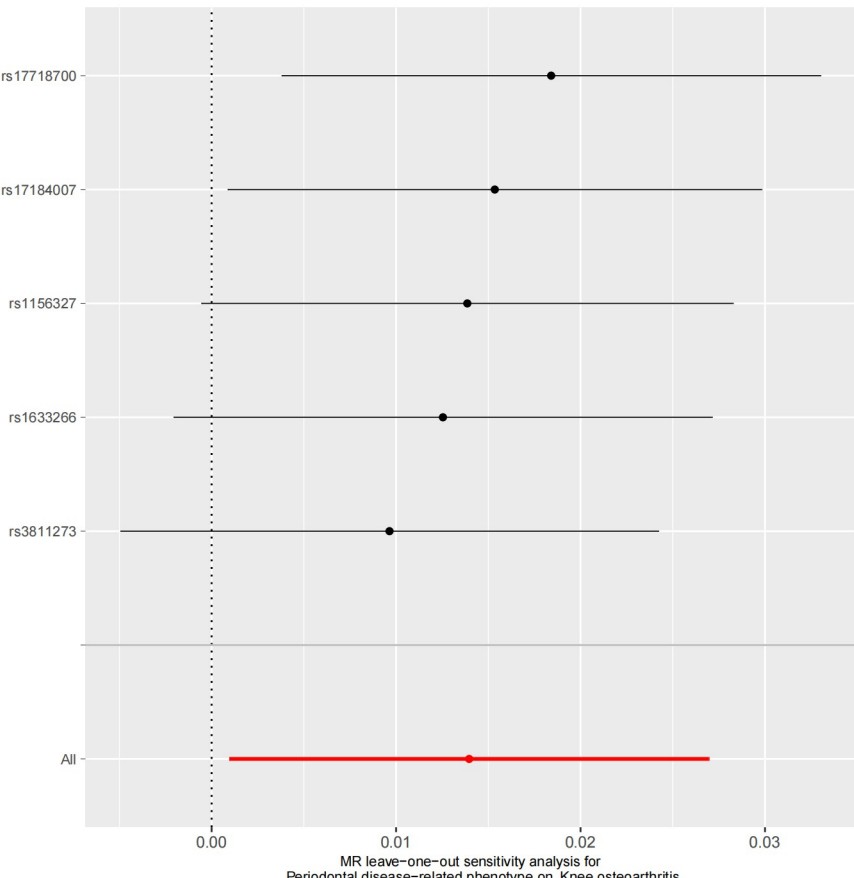

**Fig 5. Sensitivity analysis to investigate the possibility that the causal association was driven by a unique SNP in knee osteoarthritis.**

**Table 3. Heterogeneity statistics for each SNP.**

| Different exposure | MR method | Cochran Q statistic | $I^2$ | Heterogeneity P-value |
|---|---|---|---|---|
| Periodontal disease-related phenotype | MR Egger | 2.773 | 0.082 | 0.428 |
| | Inverse variance weighted | 2.945 | 0.358 | 0.567 |
| Knee osteoarthritis | MR Egger | 2.330 | 0.571 | 0.127 |
| | Inverse variance weighted | 2.370 | 0.156 | 0.306 |

**Table 4. MR estimates from each method of assessing the causal effect of KOA on the risk of PDRP.**

| Method | SNP | OR | SE | Pval | 95%CI of OR | |
|---|---|---|---|---|---|---|
| MR Egger | 3 | 297.972 | 33.15 | 0.891 | 1.80437E-26 | 4.92068E+30 |
| Weighted median | 3 | 2.965 | 1.512 | 0.472 | 0.153 | 57.427 |
| Inverse variance weighted | 3 | 3.846 | 1.244 | 0.279 | 0.336 | 44.046 |
| Simple mode | 3 | 2.335 | 1.9 | 0.699 | 0.056 | 96.747 |
| Weighted mode | 3 | 1.837 | 1.84 | 0.772 | 0.050 | 67.667 |

**Table 5. Multivariate Mendelian Randomization (MR) estimates were employed to assess the causal effect of periodontal disease-related phenotype (PDRP) and potential confounding factors on the risk of knee osteoarthritis (KOA).**

| ID | Exposure | ID | Outcome | SNP | Beta | SE | Pval |
|---|---|---|---|---|---|---|---|
| ebi-a-GCST003484 | Periodontal disease-related phenotype | ebi-a-GCST007090 | Knee osteoarthritis | 1 | -0.003 | 0.006 | 0.572 |
| ebi-a-GCST90029014 | Smoking status | | | 35 | 0.532 | 0.184 | 0.004 |
| ieu-a-1283 | Alcohol consumption | | | 1 | -0.060 | 0.179 | 0.738 |
| ukb-b-19953 | Body mass index (BMI) | | | 276 | 0.708 | 0.052 | 2.48E-42 |

**Table 6. Multivariate Mendelian Randomization (MR) estimates were employed to assess the causal effect of knee osteoarthritis (KOA) and potential confounding factors on the risk of periodontal disease-related phenotype (PDRP).**

| ID | Exposure | ID | Outcome | SNP | Beta | SE | Pval |
|---|---|---|---|---|---|---|---|
| ebi-a-**GCST007090** | Knee osteoarthritis | ebi-a-GCST003484 | Periodontal disease-related phenotype | 1 | -0.454 | 0.555 | 0.413 |
| ebi-a-GCST90029014 | Smoking status | | | 30 | 2.248 | 1.853 | 0.225 |
| ieu-a-1283 | Alcohol consumption | | | 1 | 0.228 | 1.692 | 0.893 |
| ukb-b-19953 | Body mass index (BMI) | | | 234 | -0.686 | 0.624 | 0.271 |

related phenotypes, smoking status, alcohol consumption, and BMI (Table 5). This detailed investigation provided nuanced insights into how these factors might influence KOA risk. The analysis revealed that the link between periodontal disease and KOA was not statistically significant ($\beta$ = -0.00339, SE = 0.00600, p = 0.572), suggesting that periodontal disease, as identified in previous two-sample MR analysis, may not have a direct causal impact on KOA. Conversely, we observed a significant positive association between smoking status and KOA ($\beta$ = 0.532, SE = 0.184, p = 0.0039), indicating smoking as a likely risk factor for KOA. In contrast, alcohol consumption did not show a significant causal relationship with KOA ($\beta$ = -0.0598, SE = 0.1785, p = 0.738).

In the reverse analysis, we assessed how KOA and lifestyle factors such as smoking, alcohol consumption, and BMI might affect PDRP (Table 6). Our findings suggest KOA does not have a significant impact on periodontal disease risk ($\beta$ = -0.4539, SE = 0.5550, p = 0.413). While smoking showed a potential trend towards increasing periodontal disease risk ($\beta$ = 2.248, SE = 1.853, p = 0.225), it did not reach statistical significance. No significant associations were found for alcohol consumption ($\beta$ = 0.2279, SE = 1.6917, p = 0.893) or BMI ($\beta$ = -0.6862, SE = 0.6237, p = 0.271) with the PDRP.

## Discussion

PD is increasingly recognized as a potential risk factor for KOA. Despite this, the bidirectionality of the relationship between PDRPs and KOA remains under debate. Our investigation applied five MR estimation methods, including MR-Egger, Weighted Median, IVW, Simple Mode, and Weighted Mode—to explore these associations comprehensively. Our findings reveal a consistent causal link from PDRP to KOA, particularly highlighted by the IVW method, suggesting PDRPs act as a risk factor for KOA development. Conversely, no evidence was found to suggest KOA contributes causally to PDRPs, underscoring a unidirectional relationship where PDRPs may increase KOA risk. However, when including confounders like smoking and BMI, there's a noticeable impact on KOA, particularly with BMI showing a strong association. This emphasizes the importance of accounting for lifestyle factors in the etiology of KOA.

Observational studies have noted a higher prevalence of periodontitis in patients with radiographic KOA, particularly among women, suggesting a gender-specific link [9]. Intriguingly, studies have identified common bacterial DNA, such as Porphyromonas gingivalis (Pg), in both periodontal tissue and synovial fluid of patients with arthritis [18,19]. This finding is relevant for understanding the microbial pathogenesis behind the association between PD and KOA.

Further research, including a porcine model and studies within populations with type 2 diabetes, have underscored the potential role of oral pathogens and periodontitis severity in KOA progression [20,21]. Notably, a bidirectional association between PD and KOA was observed in a Korean study, with significant findings among women [22]. A large cohort study echoed these results, revealing a bidirectional risk between OA and PD, suggesting that individuals with periodontitis are at increased risk of developing OA, and vice versa [23].

However, a contrasting MR study posited a non-causal relationship between periodontitis and arthritis, highlighting the complexity of these associations [24]. Our current study, focusing on a European population, sought to clarify the relationship between PDRP and CP in the context of KOA.

In our study, we investigated the potential causal relationships between KOA, smoking, alcohol consumption, BMI and the risk of PDRP using multivariable MR analysis. Interestingly, while smoking, alcohol consumption, and BMI [25,26] were individually identified as risk factors for PDRP in previous research, our analysis revealed no direct causal relationship between these exposures and PDRP when considered together with KOA. This suggests the presence of complex interactions or confounding factors among these variables that may influence the development of PDRP. Additionally, the lack of a significant causal effect may also indicate the need for further investigation into other potential risk factors or pathways contributing to PDRP development. Overall, our findings underscore the importance of considering multiple factors simultaneously and employing rigorous analytical methods, such as MR analysis, to better understand the complex etiology of periodontal diseases.

Reversely, we assessed the combined impact of PDRP, smoking, alcohol consumption, and BMI on KOA as the outcome. Our analysis revealed significant statistical associations between smoking and BMI with KOA risk. These findings suggest that smoking and higher BMI may independently contribute to the development of KOA. Several mechanisms could explain these associations. Smoking is known to be associated with systemic inflammation, oxidative stress, and vascular dysfunction, all of which can adversely affect joint health and contribute to the progression of osteoarthritis [27]. Similarly, elevated BMI leads to increased mechanical stress on weight-bearing joints, resulting in cartilage degradation and inflammation [28]. Moreover, results from the previous studies [29,30] also indicated a significant association between obesity-related genetic variants, smoking, and the occurrence of KOA. Additionally, adipose tissue secretes pro-inflammatory cytokines, further exacerbating joint degeneration. Therefore, our results highlight the potential importance of lifestyle factors such as smoking and maintaining a healthy weight in preventing and managing KOA.

A key strength of our study lies in the application of MR, a powerful analytical approach that leverages genetic variants as instrumental variables to investigate causal relationships while mitigating the limitations of reverse causality and confounding inherent to observational studies. By exploiting the random allocation of genetic variants during gamete formation, in accordance with Mendel's laws, MR analyses can provide more robust evidence for causal inferences. However, it is crucial to acknowledge the assumptions underlying MR and to employ a range of sensitivity analyses and statistical methods to assess and account for potential violations of these assumptions, thereby enhancing the reliability of causal estimates.

Our study is subject to several limitations that warrant careful consideration. Firstly, our exclusive focus on a European population restricts the generalizability of our findings. Genetic and environmental factors associated with periodontal disease and knee osteoarthritis may vary substantially across different ethnicities, cautioning against broad extrapolation to non-European populations. This emphasizes the necessity for similar MR studies in diverse demographic groups to validate and potentially extend our results. Moreover, the absence of detailed clinical information within the GWAS datasets hampers our ability to conduct nuanced subgroup analyses. These analyses could have provided deeper insights into how specific clinical manifestations of periodontal disease influence knee osteoarthritis risk and vice versa. The lack of granularity in the data also precludes accounting for heterogeneity within the population, such as variations in disease severity, treatment histories, or diagnostic criteria. While our study benefits from a substantial aggregate sample size, it may not fully capture the complexity of the relationship between periodontal disease and knee osteoarthritis. The intricate interplay of genetic predispositions, environmental factors, and lifestyle choices suggests that even larger, more detailed datasets could reveal nuances not observed in our analysis. Future research endeavors should strive to include more diverse populations, detailed clinical data, and comprehensive analyses of confounding factors to advance our understanding of the causal links between periodontal disease and knee osteoarthritis, facilitating the development of targeted prevention and management strategies for these conditions. In conclusion, our Mendelian randomization study provides evidence for a unidirectional causal effect of periodontal disease-related phenotype on the risk of knee osteoarthritis in the European population studied. However, after accounting for potential confounders like smoking and BMI, the direct causal relationship was attenuated, highlighting the complex interplay between these factors in the etiology of KOA. Further investigation into the shared pathways, common risk factors, and potential bidirectional effects between these conditions is warranted, emphasizing the importance of a multifaceted approach to prevention and management strategies.

## Author Contributions

**Conceptualization:** Longqiang Shen.

**Formal analysis:** Longqiang Shen.

**Funding acquisition:** Longqiang Shen.

**Supervision:** Gang Deng.

**Writing – original draft:** Di Niu.

**Writing – review & editing:** Gang Deng.

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
