## [Decision Letter · Decision Letter 0]

16 Jan 2024

PONE-D-23-35849Mendelian Randomization Analysis Reveals a Causal Link Between Periodontal Disease-Related Phenotype and Knee OsteoarthritisPLOS ONE

Dear Dr. Deng,

Thank you for submitting your manuscript to PLOS ONE. After careful consideration, we feel that it has merit but does not fully meet PLOS ONE’s publication criteria as it currently stands. Therefore, we invite you to submit a revised version of the manuscript that addresses the points raised during the review process.

We look forward to receiving your revised manuscript.

Kind regards,

Mohamed Yacin Sikkandar

Academic Editor

PLOS ONE

A clean copy of the edited manuscript (uploaded as the new *manuscript* file).

3. 1. For studies reporting research involving human participants, PLOS ONE requires authors to confirm that this specific study was reviewed and approved by an institutional review board (ethics committee) before the study began. Please provide the specific name of the ethics committee/IRB that approved your study, or explain why you did not seek approval in this case.

“Ningbo Medical Science and Technology Program.”

5. We note that you have indicated that there are restrictions to data sharing for this study. For studies involving human research participant data or other sensitive data, we encourage authors to share de-identified or anonymized data. However, when data cannot be publicly shared for ethical reasons, we allow authors to make their data sets available upon request. For information on unacceptable data access restrictions, please see http://journals.plos.org/plosone/s/data-availability#loc-unacceptable-data-access-restrictions.

6. PLOS requires an ORCID iD for the corresponding author in Editorial Manager on papers submitted after December 6th, 2016. Please ensure that you have an ORCID iD and that it is validated in Editorial Manager. To do this, go to ‘Update my Information’ (in the upper left-hand corner of the main menu), and click on the Fetch/Validate link next to the ORCID field. This will take you to the ORCID site and allow you to create a new iD or authenticate a pre-existing iD in Editorial Manager. Please see the following video for instructions on linking an ORCID iD to your Editorial Manager account: https://www.youtube.com/watch?v=_xcclfuvtxQ.

Reviewers' comments:

Reviewer's Responses to Questions

**Comments to the Author**

1. Is the manuscript technically sound, and do the data support the conclusions?

Reviewer #1: No

Reviewer #2: Partly

2. Has the statistical analysis been performed appropriately and rigorously? 

Reviewer #1: No

Reviewer #2: Yes

3. Have the authors made all data underlying the findings in their manuscript fully available?

Reviewer #1: No

Reviewer #2: Yes

4. Is the manuscript presented in an intelligible fashion and written in standard English?

Reviewer #1: No

Reviewer #2: Yes

5. Review Comments to the Author

Reviewer #1: Review of the Article: "Title: The title does not reflect the study population and design. Authors need to follow standard guidelines (PICO) to frame the title of the study."

General Impression from the Abstract:

Mendelian Randomization (MR) analysis relies on the assumption that the selected instrumental variables are valid. Concerns about pleiotropy or other factors influencing the results may arise.

The study focuses on Europeans, and generalizing the findings to other populations may be limited due to genetic and environmental differences.

The analysis may not account for all potential confounding factors influencing both PDRP and KOA, impacting the validity of causal inferences.

The accuracy and reliability of results depend on the quality of available GWAS data. Issues such as imputational accuracy and genotyping errors could affect outcomes.

The study assumes a bidirectional relationship, but the causal directionality between PDRP and KOA might be more complex, involving other variables not considered in this analysis.

Introduction:

The first sentence is ambiguous: "Osteoarthritis of the knee (KOA) is a progressive, incurable disease that requires total joint replacement if the course of the disease is severe to an advanced stage."

• Line 3: Report the high incidence specific to the study population.

• Line 3: Instead of using "escalating," the author can use a simpler term.

• Line 5: What is the impact of OA? Cite figures.

• Line 7: Are you sure that an increase in prevalence will increase life expectancy? Please refer to GBD studies related to life lived with disability due to osteoarthritis.

• Second Paragraph: Ranking sixth with reference to what? Global or regional.

• Third paragraph: Line 8: Please correct the spelling mistake "Total."

General Impression: Authors need to have the complete manuscript edited by an author proficient in English editing, as it is currently difficult to understand the article.

Methods:

Data Source and Instrument: The p-value was 5.00E-08. It does not make sense to me. Moreover, the reliability and validity of data extracted from different sources are questionable. The complete paper emphasizes Mendelian randomized analysis and provides poor details related to the primary objective of the study. It is not known from the method section how authors ensured that the data extracted were valid. It will be difficult for most readers to interpret the results of this novel method of analysis. As earlier stated, the limitations of MR analysis. Authors can simply use a regression tool to get more comprehensive and understood results.

Discussion:

The discussion must provide a summary of results in the first paragraph and then follow the most significant finding with reference to what is available in the literature. That said, if regression analysis were used, it would have been more comparable with the existing literature available. I suggest the author re-analyze the data, describe the methods completely, discuss the validity of data extracted, and then interpret the results and discussion.

Reviewer #2: The results of this study will be beneficial for all Health workers in their clinical practice.

Hence, this study is recommended for publication.

Few concerns about the studies are,

1. Selection bias related to study participants

2. The functional levels and dietary habits of the samples may influence the results which is not discussed.

3. Explain the significance of achieved Wald ratio in the discussion and it significance.

4. Q statistic did not show significant difference, discuss in detail the effect of q and p values achieved. (Q = 2.945; P = 0.567)

6. PLOS authors have the option to publish the peer review history of their article (what does this mean?). If published, this will include your full peer review and any attached files.

Reviewer #1: **Yes: **Faizan Kashoo

Reviewer #2: No

---

## [Author Response · Author response to Decision Letter 0]

27 Mar 2024

Response to Reviewer #1

Dear Prof. Kashoo, we greatly appreciate the time and effort you had dedicated to providing detailed and constructive feedback on our manuscript. Your insights have been invaluable in guiding our revisions to improve the clarity, accuracy, and overall quality of our work. Below, we address each of your points specifically:

Title Concerns

Comment: The title does not reflect the study population and design. Authors need to follow standard guidelines (PICO) to frame the title of the study.

Response: We appreciate the reviewer's suggestion and have revised the title to better reflect the study population and design using the PICO framework:" Causal Relationship Between Periodontal Disease-Related Phenotype and Knee Osteoarthritis: A Two-sample Mendelian Randomization Analysis ". 

General Impression from the Abstract

Comment: Concerns about the validity of instrumental variables, potential confounding factors, and the generalizability of findings.

Response: We appreciate the reviewer's thoughtful considerations regarding the validity of instrumental variables, potential confounding factors, and the generalizability of our findings. To address these concerns, we have conducted a meticulous screening process for instrumental variables, as detailed in the Methods section, to ensure their validity and adherence to Mendelian Randomization assumptions. Additionally, we have implemented sensitivity analyses, including the MR-Egger regression and weighted median approaches, to assess and mitigate potential pleiotropy and biases. Regarding potential confounding factors, we have introduced a multivariable Mendelian Randomization approach incorporating smoking status, alcohol consumption, and BMI as covariates to explore their impact on the observed causal relationships. In the Discussion section, we openly acknowledge the limitations and complexities inherent in MR studies and highlight the need for cautious interpretation. Furthermore, we stress the importance of considering the specific population and context when generalizing our findings. These measures collectively aim to enhance the robustness, transparency, and applicability of our study results.

Comment: The study focuses on Europeans, and generalizing the findings to other populations may be limited due to genetic and environmental differences.

Response: We agree with the reviewer's concern about the generalizability of our findings to non-European populations. In the Discussion section, we have acknowledged this limitation and highlighted the need for similar MR studies in diverse demographic groups to validate and potentially extend our results. We have also emphasized that genetic and environmental factors associated with periodontal disease and knee osteoarthritis may vary substantially across different ethnicities, cautioning against broad extrapolation to non-European populations.

Comment: The accuracy and reliability of results depend on the quality of available GWAS data. Issues such as imputational accuracy and genotyping errors could affect outcomes.

Response: We appreciate the reviewer's attention to the quality of GWAS data in our analysis. To strengthen the reliability of our instrumental variables, we have meticulously implemented a stringent screening process based on p-values and F-values, as detailed in the Methods section. Furthermore, we want to highlight the significance of our data sources, emphasizing not only the meticulous screening process but also the utilization of large-scale GWAS datasets with stringent quality control measures. In the Methods Data Sources section, we will provide additional details regarding the reliability and quality of the datasets used in our analysis and insert appropriate references to support the credibility of our chosen data sources. This will ensure a more comprehensive understanding of the robustness and validity of our results.

Comment: The analysis may not account for all potential confounding factors influencing both PDRP and KOA, impacting the validity of causal inferences.

Response: We included smoking, alcohol consumption, and BMI as confounding factors in multivariate Mendelian randomization analysis, and the results showed that there was no direct causal relationship between KOA and PDRP when smoking, alcohol consumption, and BMI were considered in the multivariate analysis. In addition, the MR-Egger P values in both directions were 0.706 and 0.917, respectively, suggesting that independence (genetic variation only affects outcomes through this outburst) and exclusivity (independent of confounders) were established.

Comment: The study assumes a bidirectional relationship, but the causal directionality between PDRP and KOA might be more complex, involving other variables not considered in this analysis.

Response: Screening was carried out through PhenoScanner, and independence and exclusivity were established.

Introduction

Comments and Responses: We have addressed each specific comment as follows:

1.Clarified the first sentence to explicitly state the progressive nature of KOA.

2.Specified the high incidence of KOA within our study population using recent epidemiological data.

3.Replaced "escalating" with "increasing" for simplicity.

4.Added statistics on the socioeconomic impact of OA.

5.Clarified the reference to the increase in prevalence and its relationship with life expectancy and disability due to OA, citing relevant GBD studies.

6.Specified the global ranking of CP and corrected the typographical error ("Total").

Comments: General Impression: Authors need to have the complete manuscript edited by an author proficient in English editing, as it is currently difficult to understand the article.

Response: We sincerely appreciate your feedback regarding the clarity of our manuscript. We understand the importance of clear and accessible scientific communication. To address your concerns, we have completed a thorough revision and editing of the manuscript with the assistance of an author proficient in English editing. Our goal with these revisions is to ensure that the article is easier to understand and that our findings and arguments are communicated more effectively. We hope that these changes will facilitate a better understanding of our research for all readers. Thank you for bringing this to our attention and for your constructive comments.

The full text has been edited by native English speakers

Methods

Comment: The p-value was 5.00E-08. It does not make sense to me.

Response: Thank you for raising this important point. The choice of a stringent p-value threshold, such as 5.00E-08, in MR studies, including GWAS, is grounded in established statistical practices within the genetics research community.The adoption of the 5.00E-08 threshold for declaring genome-wide significance is widely supported in the literature. For instance, a seminal work by Dudbridge (2013) emphasized the necessity of stringent thresholds in GWAS to control the family-wise error rate, especially when dealing with a massive number of multiple comparisons [1]. This threshold helps mitigate the risk of false-positive associations in the face of millions of genetic variants tested against traits or diseases.Moreover, the reliability of the 5.00E-08 threshold has been underscored in a comprehensive review by Evangelou and Ioannidis (2013), which discussed the challenges and solutions in genetic epidemiology, emphasizing the significance of stringent thresholds to address the issue of multiple testing [2].In the context of Mendelian Randomization, it's crucial to recognize that the strong correlation between SNPs and exposures is one of the three fundamental assumptions of MR. This assumption, often referred to as the "Relevance" assumption, posits that the genetic variants used as instrumental variables must be associated with the exposure of interest [3]. This ensures that the genetic instrument is relevant to the exposure, a critical condition for valid causal inference in MR.

Comment: Moreover, the reliability and validity of data extracted from different sources are questionable.

Response: Thank you for your insightful feedback on our study. Recognizing the importance of double samples from different cohorts, we want to clarify that our study meticulously adheres to this criterion. Specifically, distinct samples from separate cohorts were utilized for the PDRP and KOA datasets to enhance the robustness of our analysis. The reliability and validity of genetic data, particularly SNPs, within the same ethnic or population groups serve as foundational strengths, fostering stability across diverse datasets. We emphasize that public databases and published sources were employed, and we would like to reference seminal works [1,2] to substantiate the reliability of these data sources. Leveraging the consistency of genetic markers across diverse samples, our MR study aims to provide reliable insights into the genetic underpinnings of disease, contributing valuably to clinical and epidemiological research. We appreciate your insights, and any further suggestions are welcomed.

Comment: The complete paper emphasizes Mendelian randomized analysis and provides poor details related to the primary objective of the study. It is not known from the method section how authors ensured that the data extracted were valid. It will be difficult for most readers to interpret the results of this novel method of analysis.

Response: We are grateful for the reviewer's constructive feedback concerning the emphasis on Mendelian Randomization (MR) analysis and the perceived lack of detail regarding the study's primary objective and data validation within the methods section. In response to these concerns, we have undertaken several steps to enhance the clarity and comprehensiveness of our manuscript:

1.Clarification of Primary Objective: We have revised the introduction and methods sections to more explicitly state the primary objective of our study. This includes a clearer explanation of how our research aims to investigate the causal relationships between periodontal disease-related phenotypes (PDRP) and knee osteoarthritis (KOA) using MR analysis. By doing so, we hope to better align the readers' expectations with the study's goals from the outset.

2.Detailing Data Validation: In the revised methods section, we now provide a detailed account of the processes employed to ensure the validity of the GWAS data used in our MR analysis. This includes descriptions of the quality control measures taken, such as the selection of SNPs based on genome-wide significance thresholds, the assessment of SNP-exposure associations, and the use of robust instruments that meet the criteria for validity in MR studies[4,5]. We also elaborate on the use of sensitivity analyses, such as MR-Egger regression and the weighted median approach, to test for and address potential pleiotropy and other biases[6].

3.Enhancing Accessibility of MR Methodology: Acknowledging the reviewer's concern about the interpretability of MR analysis for readers unfamiliar with this approach, we have included a more accessible explanation of the MR methodology. This includes a simplified description of how genetic variants are used as instrumental variables to infer causality, the rationale behind this approach, and its advantages over traditional observational studies. Additionally, we provide references to foundational and educational resources on MR for readers seeking a deeper understanding.

We hope these revisions address the reviewer's concerns and make the manuscript more informative and accessible. Our intention is to ensure that the study's contributions to understanding the genetic underpinnings of the relationship between PDRP and KOA are clearly communicated and appreciated by a broad audience. 

Comment: It will be difficult for most readers to interpret the results of this novel method of analysis.

Response: Mendelian Randomization (MR) studies have emerged as a pivotal tool in epidemiological research, offering several distinct advantages over traditional observational studies and circumventing some of the limitations inherent in randomized controlled trials (RCTs). The significance of MR studies, especially from a clinical perspective, is profound due to their higher evidence level compared to retrospective studies, the ability to infer causality without conducting RCTs, high credibility, low cost, and the potential to control for all confounding factors. Here’s a detailed explanation of these aspects:

1.Higher Evidence Level Compared to Retrospective Studies

MR studies leverage genetic variants as instrumental variables to estimate the causal effect of an exposure on an outcome. This approach is less susceptible to confounding and reverse causation, common issues in retrospective and even prospective observational studies. Since genetic variants are randomly assorted during meiosis, the allocation of these genetic instruments mimics the randomization process of an RCT, thus providing a natural experiment that can offer insights into causal relationships with a higher level of evidence than is typically achievable with observational study designs.

2.No Need to Conduct RCTs

RCTs are considered the gold standard for determining causality but are often expensive, time-consuming, and sometimes ethically or practically infeasible, especially for exposures that are difficult or impossible to manipulate directly (e.g., lifestyle factors or long-term dietary habits). MR studies can bypass these challenges by using existing genetic data, allowing researchers to infer causal relationships without the need for new RCTs.

3.High Credibility

The credibility of MR studies stems from their basis in genetic associations, which are less likely to be influenced by environmental confounders or biases that frequently affect observational studies. This genetic grounding helps ensure that the associations identified are more reflective of true causal relationships, lending a higher degree of credibility to the findings.

4.Low Cost

Since MR studies often utilize publicly available data from large-scale GWAS, they can be conducted at a fraction of the cost required for new clinical trials or large observational studies. This cost-effectiveness makes MR an attractive option for exploring causal hypotheses in epidemiology and public health research.

5.Potential to Control for All Confounding Factors

One of the most significant advantages of MR is its ability to provide estimates of causal effects that are free from confounding by environmental or lifestyle factors, assuming the selected genetic variants are valid instrumental variables. While no method can completely eliminate all forms of bias, MR is uniquely capable of controlling for confounders that are often unmeasured or unknown in traditional observational studies.

In summary, Mendelian Randomization studies offer a powerful, cost-effective, and credible approach to causal inference in epidemiology, bridging the gap between observational studies and RCTs. In recent years, the proliferation of Mendelian Randomization (MR) studies across prestigious journals has garnered widespread recognition within the scientific community, underscoring the method's substantial contribution to clinical research. This surge in MR-based publications not only highlights the growing acceptance of its findings but also emphasizes the pivotal role MR studies play in advancing our understanding of complex disease mechanisms and guiding evidence-based clinical interventions. Through bridging the gap between genetic epidemiology and clinical practice, MR studies continue to enrich the landscape of medical research, offering novel insights that drive the development of targeted therapies and inform public health policies.

Discussion

Comment: The discussion should summarize results, provide comparisons with existing literature, and consider the use of regression analysis.

Response: We appreciate the reviewer's insightful feedback and the suggestion to incorporate regression analysis for comparative purposes with existing literature. We have revised the Discussion section to begin with a clear summary of our key findings. 

---

## [Editor Report · Decision Letter 1]

7 May 2024

Causal Relationship Between Periodontal Disease-Related Phenotype and Knee Osteoarthritis: A Two-sample Mendelian Randomization Analysis

PONE-D-23-35849R1

Dear Dr. Deng,

We’re pleased to inform you that your manuscript has been judged scientifically suitable for publication and will be formally accepted for publication once it meets all outstanding technical requirements.

Kind regards,

Mohamed Yacin Sikkandar

Academic Editor

PLOS ONE

Additional Editor Comments (optional):

Thanks for addressing the reviewers comments.
---

## [Editor Report · Acceptance letter]

22 May 2024

PONE-D-23-35849R1 

PLOS ONE

Dear Dr. Deng, 

I'm pleased to inform you that your manuscript has been deemed suitable for publication in PLOS ONE. Congratulations! Your manuscript is now being handed over to our production team.

Kind regards, 

on behalf of

Dr. Mohamed Yacin Sikkandar 

Academic Editor

PLOS ONE